# Quantitative Assessment of Body Composition in Cirrhosis

**DOI:** 10.3390/diagnostics14192191

**Published:** 2024-09-30

**Authors:** Christian Skou Eriksen, Søren Møller

**Affiliations:** 1Department of Clinical Physiology and Nuclear Medicine, Center for Functional and Diagnostic Imaging and Research, Hvidovre Hospital, 2650 Hvidovre, Denmark; soeren.moeller@regionh.dk; 2Institute of Clinical Medicine, Faculty of Health Sciences, University of Copenhagen, 2200 Copenhagen, Denmark

**Keywords:** cirrhosis, body composition, quantitative imaging, sarcopenia, myosteatosis, fat distribution

## Abstract

Changes in body composition often accompany the progression of liver disease and seem to be an aggravating pathophysiological factor. Specifically, accelerated loss of skeletal muscle mass, lower muscle quality, and changes in body fat distribution have been shown to be associated with poor clinical outcomes. The aim of the present narrative review was to discuss the current status and relevance of commonly applied, advanced, non-invasive methods to quantify skeletal muscle mass, muscle fat infiltration—i.e., myosteatosis—and fat distribution. This review focuses in particular on Computed Tomography (CT), Dual-energy X-ray Absorptiometry (DXA), Bioelectrical Impedance Analysis (BIA), Magnetic Resonance Imaging (MRI), and Ultrasonography (US). We propose future directions to enhance the diagnostic and prognostic relevance of using these methods for quantitative body composition assessment in patients with cirrhosis.

## 1. Introduction

Accelerated loss of skeletal muscle mass and function, also termed sarcopenia, is an aggravating pathophysiological comorbidity to cirrhosis. Sarcopenia increases the risk of falls and fractures, impairs activities of daily living, lowers quality of life, and increases the risk of cirrhosis-related organ dysfunction and death [1,2,3,4].

The prevalence of sarcopenia in liver disease has been estimated to be 33–38% [2,5], which is higher than in the general aging population, and high compared to most other chronic diseases [6]. Recent clinical practice guidelines recommend early diagnosis of sarcopenia in chronic liver disease [7,8,9] in order to accommodate associated complications. Early diagnosis could potentially promote nutritional and exercise interventions and optimize medical treatment, including anabolic therapy. An additional clinical implication of diagnosing sarcopenia is improved prognostic classification, which would contribute to the prioritization of surgical intervention. According to the evidence-based guidelines for diagnosing sarcopenia in cirrhosis [10,11,12,13], advanced techniques for precise and valid quantification of skeletal muscle should be applied.

Evidence is emerging that fat infiltration in skeletal muscle (myosteatosis) and changes in fat distribution expressed as a high visceral-to-subcutaneous fat ratio are poor prognostic markers in patients with cirrhosis [14,15]. Assessment of fat distribution may also become relevant in the fast-growing population of patients with metabolic-associated steatotic liver disease (MASLD) [16]. Currently, there is no consensus about how to diagnose a pathological fat distribution.

The pathophysiological processes linking skeletal muscle loss and altered fat distribution to liver disease are complex and not fully understood. Physical inactivity and suboptimal nutrition associated with critical disease partially explain the association. Several experts in the field of cirrhosis also argue for a vicious liver–muscle–fat cycle involving negative metabolic, immunological, and endocrine interactions between the tissues [17,18,19] (Figure 1).

Although assessment of muscle strength and physical performance is recommended as the cornerstone of the diagnosis of sarcopenia, the advancement of non-invasive imaging techniques accelerates the use of skeletal muscle mass assessments. In patients with musculoskeletal impairments or acutely ill patients, muscle mass assessment may even be the primary or only valid evaluation of skeletal muscle. The advanced techniques also make it possible to quantify fat compartments accurately with increasing diagnostic and prognostic utility.

This narrative review discusses the status and relevance of advanced methods for non-invasive quantitative body composition measurement in patients with cirrhosis with an emphasis on skeletal muscle mass and fat distribution. We propose potential directions for future research.

## 2. Materials and Methods

This study was designed as a narrative review. A literature search was performed on pubmed.gov with the search items listed below (search items in title or abstract). The search was limited to human studies from the years 2018–2023.

(“Computed tomography” OR “CT”) AND (“sarcopenia” OR “muscle” OR “myosteatosis” OR “fat” OR “adipose”) AND (“cirrhosis” OR “liver disease”)

(“Dual-energy X-ray absorptiometry” OR “DXA” OR “DEXA”) AND (“lean” OR “muscle” OR “sarcopenia” OR “fat” OR “adipose”) AND (“cirrhosis” OR “liver disease”)

(“Bioelectrical impedance” OR “BIA” OR “Phase angle”) AND (“sarcopenia” OR “muscle” OR “lean” OR “fat” OR “adipose”) AND (“cirrhosis” OR “liver disease”)

(“MRI” OR “magnetic resonance imaging”) AND (“sarcopenia” OR “muscle” OR “myosteatosis” OR “fat” OR “adipose”) AND (“cirrhosis” OR “liver disease”)

(“Ultrasound” OR “ultrasonography”) AND (“sarcopenia” OR “muscle” OR “fat” OR “adipose”) AND (“cirrhosis” OR “liver disease”)

Titles and abstracts were screened for relevance. Reference lists were screened for additional papers of relevance. References to the literature published earlier than 2018 were included in some cases when it was relevant.

## 3. Diagnostic Aspects of Sarcopenia in Cirrhosis

The term sarcopenia was first used in the beginning of the 1990s to describe pathologically low muscle mass in aging individuals [20]. Since then, several operational definitions of sarcopenia have been published. Among others, the International Working Group on Sarcopenia (IWGS) [21], the Foundation for the National Institutes of Health (FNIH) [22], the European Working Group of Sarcopenia in Older People (EWGSOP2) [23], the Asian Working Group on Sarcopenia (AWGS) [8], and the Sarcopenia Definition and Outcomes Consortium (SDOC) [24]. Specific criteria for sarcopenia in patients with cirrhosis have been published by the Japanese Society of Hepatology (JSH) [11], the European Association for the study of liver diseases (EASL) [12], the Indian National Association for study of the liver (INASL) [13], the American Association for the Study of Liver diseases (AASLD) [10], and the European Society for Clinical Nutrition and Metabolism (ESPEN) [25]. The reports reveal some differences as to the modalities, variables, and cut-offs to apply, which may to some extent be explained by ethnic differences in body composition, different availability of equipment, and local traditions. For instance, the prevalence of sarcopenia is generally higher in Asian populations when using cut-offs from North American/European populations [13,26]. Regional population-based methods and cut-offs may therefore give higher precision in prognostic models [27].

The EWGSOP2 provides an operational flow-chart to diagnose sarcopenia, including (1) a questionnaire screening to find patients at risk, (2) the measurement of muscle strength, (3) confirmation with the measurement of muscle mass, and (4) physical performance tests, to categorize the severity of sarcopenia [23]. Although we focus on advanced muscle mass quantification, we acknowledge that the use of questionnaires together with the assessment of muscle strength and physical performance, i.e., a hand grip strength test, chair stand test, and/or gait speed, are the primary tests in the diagnosis, as well as in the continued clinical evaluation of sarcopenia in patients with liver disease. Table 1 summarizes recommendations for the measurement of skeletal muscle mass from the different consensus groups.

### 3.1. Assessment of Muscle Mass

The measurement of muscle mass is used to confirm the diagnosis of sarcopenia and is also relevant as a primary evaluation of skeletal muscle status in patients where strength testing is compromised due to arthritis, neuromuscular disease, or acute illness. Additionally, the measurement of muscle mass is not dependent on cooperation between assessor and patient, and some liver centers may not have trained staff or clinical routine in performing strength tests.

Accurate measurement of skeletal muscle mass requires the use of advanced 4-compartment models, which are often invasive and time-consuming [28,29,30,31]. Simpler and faster non-invasive methods can be applied in clinical settings with reasonable accuracy and reproducibility. The most frequently reported methods are Computed Tomography (CT), Bioelectrical Impedance (BIA), and Dual-energy X-ray Absorptiometry (DXA), which are also the methods recommended by the clinical practice guidelines [8,11,22,23]. Other non-invasive methods used to quantify skeletal muscle are magnetic resonance imaging (MRI) and ultrasonography (US), which will also be mentioned below. Anthropometric methods to quantify skeletal muscle are not covered here.

#### 3.1.1. Computed Tomography (CT)

CT produces grayscale images based on different attenuations of X-rays in tissues of different densities. The unit of measurement is Hounsfield Units (HU), which is calibrated to air (−1000) and water (0), with higher values reflecting more dense tissue. The commonly applied HU interval for skeletal muscle is −29 to +150 HU [9,32].

CT has high precision and accuracy and, together with MRI, is considered to be a non-invasive gold-standard technique to assess skeletal muscle mass [32]. The major disadvantage of CT is radiation exposure, which limits its use for repeated assessments. Expenses and availability are also a limitation in some regions of the world (Table 2).

CT has been used for quantitative measurement of skeletal muscle mass for at least three decades in different populations and patient groups [32,33,34]. The most common measure in cirrhosis is the skeletal muscle area in the axial plane at the level of the third vertebra normalized to height squared, also called the skeletal muscle index (L3-SMI) [5]. Although a few studies have found no association between L3-SMI and mortality [35], there is general agreement that low L3-SMI predicts morbidity and mortality in advanced liver disease [36,37,38,39,40,41], and this is reflected by the recommendation to use this measure by consensus groups in hepatology from across the world. Interestingly, the recommended cut-off for L3-SMI in men diverges between Asian and Japanese populations (L3-SMI < 42 kg/m^2^) versus North American and European populations (L3-SMI < 50 kg/m^2^), which likely reflects natural differences in stature and physiology. The recommendation to use L3-SMI based on CT-scans differs from the recommendations for healthy older adults, where CT is considered a secondary option after DXA and BIA. The decision to use CT in patients with cirrhosis may be advantageous due to less influence of hydration on the muscle mass estimate compared to DXA and BIA (discussed below). Additionally, CT is often performed for other diagnostic purposes in patients with cirrhosis, making these available for secondary analysis of muscle mass.

There is some evidence that surgical insertion of a transjugular portosystemic shunt (TIPS) increases L3-SMI and that this is associated with better survival [42,43,44], whereas others have reported no improvement of L3-SMI after liver transplantation [45]. The beneficial effect of receiving a healthy liver on skeletal muscle may be counterbalanced by other post-transplantation factors that negatively affect skeletal muscle such as immunosuppressive treatment and general effects of major surgery. More knowledge is needed about how changes in L3-SMI over the course of disease development and after different treatments relate to changes in morbidity and mortality.

Measurement of transverse psoas muscle area on CT normalized to height squared (PMI) is another frequently reported measure of skeletal muscle mass, which has shown good prognostic performance in relation to time to re-hospitalization [46] and mortality [47,48,49]. Also, PMI has been reported to increase after TIPS [50]. A recent study calculated the psoas muscle volume index, which had a superior association with mortality than the simple PMI based on CSA [51]. Another study calculated the psoas muscle depletion index based on expected psoas muscle mass from healthy donors and concluded that this model was less confounded by other measures of body stature such as height and weight [52]. It is possible that advanced algorithms for muscle mass assessment are more accurate than simple CSA measurements, and that the method of assessment is more important than which muscle group is assessed. Volumetric assessment may become more easily accessible with the advancement of AI technology.

The lower erector spinae muscle has also been shown to be a valid measure of skeletal muscle status in patients with cirrhosis [53], and other spinal levels may also be relevant locations for skeletal muscle assessment, especially in cases where the L3 level is not involved in the available scan area. More research is needed to confirm this.

There are only a few reports on the measurement of leg skeletal muscle, although this may be a relevant measurement for functional performance. Specialized CT equipment (peripheral quantitative CT (pQCT)) offers the possibility of quantitative skeletal muscle analysis of peripheral muscles with far less radiation. Analysis of mid-thigh muscle CSA with pQCT is a robust method for skeletal muscle assessment, and specific cut-offs have been proposed for diagnosing sarcopenia in older adults [54,55]. To our knowledge, there are no data on patients with cirrhosis. Low access to equipment is probably the main reason that this method has not gained more interest in patients with cirrhosis.

Identification and drawing of specific regions of interest (ROIs) on axial CT slices have previously been cumbersome and operator-dependent, but the implementation of completely or partially automated measurements and artificial intelligence (AI) technology has changed this [41,56,57,58,59]. In the future, muscle mass will likely be automatically reported and included in advanced disease status models. This will potentially lead to a huge increase in data availability, which may help validate the selection of the best site and method for muscle assessment and specific cut-offs. CT scans thus seem an obvious first choice for muscle mass assessment in liver disease, when the scans are already available for other diagnostic purposes. However, more knowledge is needed about how changes in skeletal muscle mass over the time course of disease development and after different interventions change morbidity and mortality. A limitation of CT is the high radiation compared to other methods, which makes the method less useful for repeated measurements, especially in patients with a long life expectancy. Other less expensive and radiation-free methods may be better suited for this purpose.

#### 3.1.2. Dual-Energy X-ray Absorptiometry (DXA)

DXA uses X-rays with two different energy levels to separate the body into osseous and soft tissue. The soft tissue can be subdivided into fat and lean tissue. Lean tissue consists of skeletal muscle, organs, and extracellular fluid. There are different validated equations for the estimation of skeletal muscle mass based on DXA lean mass measurement [60]. Although one study showed that DXA may underdiagnose sarcopenia [61], DXA generally has high precision and accuracy compared to invasive 4-compartment models, including dilution of deuterated water [28,31] and non-invasive gold-standard techniques like CT [62]. Further, DXA has a relatively low cost compared to CT and MRI, and DXA exposes the patient to far less radiation than CT, which makes it suitable for repeated measurements (Table 2).

Compared to CT, there are fewer studies that have used DXA to evaluate skeletal muscle mass in patients with cirrhosis. Most studies report a prevalence of sarcopenia with DXA that is in the same range as when determined by CT, and DXA-determined sarcopenia increases mortality risk in advanced liver disease of mixed etiologies [63,64,65], as well as in earlier stages of MASLD [66]. Low muscle mass has also been associated with increased liver fibrosis even after adjusting for known confounding socioeconomic and medical conditions, suggesting that low muscle mass increases the risk of developing MASLD [67], underlining the reciprocal relation between skeletal muscle and the liver (Figure 1).

Most international working groups on sarcopenia in healthy older adults recommend the use of the appendicular skeletal muscle index (ASMI) for confirmation of sarcopenia (Table 1). ASMI is the sum of arm and leg lean mass normalized to height squared [23]. ASMI correlates well with whole-body skeletal muscle mass measured with MRI [68]. The FNIH recommends adjustment of appendicular skeletal muscle to body mass index (BMI) instead of height [22] in order to accommodate obese patients, which may be relevant in the fast-growing population of patients with MASLD. One study also found that appendicular lean mass adjusted to weight and BMI, but not to body height, was associated with a higher prevalence of MASLD and liver fibrosis [67]. Correction to BMI can, however, also lead to overcorrection in patients with fluid overload due to decompensation, which calls for different adjustment methods in different patient groups. In contrast to the European and Asian recommendations, the SDOC recommends not using DXA lean mass for diagnosing sarcopenia due to the weak correlation to strength measurements [24].

The liver disease-specific practice guidelines generally recommend CT L3-SMI instead of DXA lean mass. The EASL and INASL recommend DXA-ASMI as an alternative to CT with reference to cut-offs for healthy older adults (Table 1), whereas The AASL does not recommend DXA to confirm sarcopenia in liver disease. To our knowledge, there are no reported cirrhosis-specific cut-offs for DXA-ASMI. The lack of specific cut-offs and the reluctance to recommend DXA in patients with cirrhosis may be due to the challenge with hydration status. Changes in the water content of visceral organs bias DXA lean mass estimation is especially relevant in patients with decompensated cirrhosis with ascites and peripheral edema. One study measured ASMI in patients with cirrhosis before and after paracentesis and found it to be unaffected by the presence of ascites [65]. However, peripheral edema may still contribute significantly to arm and leg lean mass estimation. Recent studies suggest that especially leg lean mass may be overestimated in patients with cirrhosis, and that arm lean mass is closely associated with worse outcomes than both leg lean mass and ASMI [63,64,69,70]. Surprisingly, this only seems to be the case for men [63,71], but the reason for this gender difference is unknown. Thus, arm lean mass may be a better prognostic marker in patients with cirrhosis, especially in decompensated patients. Cut-offs for low arm lean mass remain to be validated.

Although DXA-ASMI is a validated and cost-effective method to diagnose sarcopenia, it is not recommended for routine analysis of skeletal muscle in patients with liver disease. The challenge with fluid retention and easy access to CT scans (already available) as well as low access to and experience with DXA are possible explanations. However, using arm lean mass seems to be a valid alternative to ASMI in patients with decompensation, and is also better suited for repeated assessments than CT. ASMI may also be a valid measure of skeletal muscle mass in the early stages of liver disease. Future prospective studies should validate DXA-ASMI, as well as DXA-arm lean mass for diagnosis of sarcopenia, and determine cut-offs specific to the etiology of liver disease, as well as ethnicity and sex. Patients with MASLD are still underrepresented in studies measuring DXA lean mass and they should make up a greater proportion of the patients examined in future studies. Finally, the value of performing repeated DXA scans to track the prognostic significance of changes in muscle mass during disease development and before and after treatment remains to be investigated (Table 3).

#### 3.1.3. Bioelectrical Impedance Analysis (BIA)

Bioelectrical Impedance Analysis (BIA) uses low-frequency electrical currents of single or multiple frequencies to estimate whole-body or segmental-body composition. Based on the electrical currents, there are different prediction equations for estimating fat and skeletal muscle mass, including regional assessments (i.e., arms, legs, etc.) [72]. Phase angle, which is the ratio of resistance to reactance, is another clinically relevant BIA measure that reflects cellular health. Higher values are considered to reflect higher cellularity and membrane integrity (i.e., higher lean mass) [73]. Phase angle may be a more reproducible and valid measure of nutritional status and a better predictor of adverse outcomes compared to skeletal muscle mass estimates [73,74,75]. BIA is generally easy to use, low cost, and with no side-effects or burdens for the patient (Table 2).

BIA is a frequently reported method for body composition assessment in patients with cirrhosis and other diseases of the digestive system [76]. For the purpose of muscle mass estimation in patients with liver disease, this technique has been shown to correlate moderately with the gold standard 4-compartment model [31], as well as with CT [77,78], DXA [79], and MRI [80]. One study found BIA to be robust against changes is fluid status (i.e., ascites) [81]. However, there is evidence that the accuracy and, not least, reproducibility is compromised in patients with decompensation due to changes in fluid retention [28,31], which is the major drawback of using BIA for skeletal muscle mass assessment in cirrhosis.

Low skeletal muscle mass determined by BIA has been associated with the severity of hepatic steatosis in children with cirrhosis [82], as well as an increased risk of mortality [77]. Low phase angle has been associated with an increased risk of falls, hospitalization [83], hepatic encephalopathy [84], and mortality in several studies [75,83,85,86,87]. Two studies used only BIA arm muscle as a predictive marker since the arms are usually less affected by overhydration. Arm muscle showed better association than leg muscle to mortality [88,89], which corroborates results from DXA studies showing higher prognostic value of arm lean mass compared to leg lean mass and total lean mass [63,64,70].

There are only a few studies reporting repeated assessments of skeletal muscle mass with BIA, despite the fact that BIA, being a cheap and radiation-free method, is well suited for this purpose. One recent large prospective study in patients with MASLD measured appendicular skeletal muscle mass with BIA two times separated by 5 years [90]. They found larger reductions in skeletal muscle mass in patients with MASLD and even larger reductions in more advanced MASLD. Another recent study also used BIA for the assessment of skeletal muscle mass and total body fat before and after lifestyle and medical interventions and reported a correlation between the gain of appendicular skeletal muscle plus the loss of total fat mass and the resolution of liver steatosis [91]. A large retrospective cohort study also found a higher incidence of MASLD in patients with a decrease in skeletal muscle and an increase in fat mass over a 10-year follow-up period [92], suggesting that these body composition changes are risk factors for MASLD. These studies support the use of BIA as a valid tool for repeated assessments of body composition in patients with liver disease, but more studies are needed to confirm these findings, not least in patients with more advanced liver disease.

Several studies have calculated cut-offs for sarcopenia in different liver disease populations, with different end-points (i.e., malnutrition, mortality) and based on phase angle as well as other BIA measures of skeletal muscle mass [76]. Hence, JHS, EASL and INASL all suggest BIA lean mass normalized to body height as a reasonable alternative to CT L3-SMI, with only negligible variation in cut-offs between the different consensus groups (Table 1).

One group constructed a nomogram to assess the continued likelihood of being free of sarcopenia based on BMI and phase angle and with CT L3-SMI as a reference [93]. The model had good diagnostic accuracy and demonstrated that the diagnosis of sarcopenia could be obtained with inexpensive and easily accessible methods. Moreover, it introduced a valuable differentiation between patients with high and low BMI, since patients with high BMI are expected to have higher phase angles (relatively higher total body water) in order to rule out sarcopenia. The inclusion of BMI in the model could particularly be relevant when distinguishing patients with, i.e., alcoholic or viral etiology of cirrhosis from those with MASLD.

Collectively, BIA is a valid alternative for muscle mass estimation with low cost and low patient burden. The use of phase angle partially overcomes the challenge of fluid volume changes in patients with cirrhosis. However, rigorous comparisons to gold standard methods question the precision in repeated measurements [31], but this may be a theoretical limitation in research, which should not argue against using the method in clinical practice or large-scale research studies, when this is the most easily accessible method or when several repeated assessments are warranted.

#### 3.1.4. Magnetic Resonance Imaging (MRI)

MRI uses radiofrequency waves and a strong magnetic field to generate images of excellent soft tissue resolution without using ionizing radiation. MRI has high precision and accuracy, options for three-dimensional imaging, and provides information on both muscle and fat compartments (Table 2). There are fewer studies using MRI to quantify body composition compared to CT, probably because of higher expenses, less availability, and longer scan time. Image analysis is time-consuming but is subject to massive development in the direction of automatic reporting of quantitative image variables.

The available MRI data on sarcopenia in patients with cirrhosis adhere mainly from retrospective cohorts. The reports are similar to CT; low skeletal muscle mass on MRI is associated with worse clinical outcomes, including mortality [94,95,96,97].

Like CT, there is a variety of muscle groups chosen for the assessment of skeletal muscle mass on MRI. Abdominal MRI displays high agreement with CT for determination of skeletal muscle area [98] but is not routinely performed in the diagnostic work-up in patients with cirrhosis in all centers. Isolated liver MRI does not always include the L3 level, which has been shown to be the most optimal slice for estimation of whole-body skeletal muscle mass, at least in healthy adults [99]. A recent study aimed to circumvent this limitation by investigating the relations between skeletal muscle at the T12 to L3 spinal levels [100]. They found good diagnostic accuracy when using the more cranial spinal levels to estimate the L3 muscle area, which makes it possible to use dedicated liver MRI to quantify skeletal muscle as well. This is supported by prognostic studies, which used skeletal muscle area at the level of the superior mesenteric artery (Th12-L2) to predict worse clinical outcomes in patients with cirrhosis [94,95]. Another reported method is to use the mid-thigh quadriceps muscle area [101,102,103], but this method requires an additional scan, which makes it less suitable in a clinical setting. On the other hand, it is possible that thigh muscle is better to evaluate changes with physical inactivity and training interventions, since muscles of the extremities may be more responsive to changes in loading compared to core muscles. One group found increased muscle areas of the quadriceps femoris muscle after 3 months of resistance training and lower rates of hospitalization and death 3 years after the intervention compared to a control group [101,104], suggesting a good prognostic value of thigh muscle mass measurement.

Taken together, MRI has excellent diagnostic accuracy for the quantification of skeletal muscle without exposing the patient to radiation. Additionally, MRI has superior soft tissue spatial resolution compared to other modalities and is therefore ideal for detailed analysis of muscle quality and fat compartments, which will be discussed below. On the other hand, cut-offs for low skeletal muscle mass have not been sufficiently validated with MRI. Moreover, the currently limited availability in many parts of the world, the high cost, and time consumption make the method most applicable for small-scale research use and for validation of simpler techniques, rather than repeated clinical assessments in a growing population of patients with liver disease.

#### 3.1.5. Ultrasonography (US)

Ultrasonography uses the echo of ultrasound waves to make 2D grayscale images of the tissue. Muscle and fat have distinct echo patterns, often clearly separated by hyperechoic connective tissue fascia. The procedure is without side effects and can be performed bedside, also in patients with critical illness, or even in the patient’s own home, and the equipment is often already available for other purposes, not least the assessment of liver steatosis/fibrosis (Table 2). The widespread availability of US scanners has increased the number of studies investigating the diagnostic accuracy of this modality for skeletal muscle mass assessment. In patients with cirrhosis, US has been validated against non-invasive gold standard techniques like MRI [105] and CT [106,107,108]. However, there is evidence that US in patients with cirrhosis is still less accurate than CT, MRI, DXA and BIA compared to an invasive gold-standard technique [31]. One reason may be that ultrasound is highly dependent on the operator and lacks standardization. Differences in contrast settings, frequency, measurement location, measurement angle, application of pressure, choice of probe (curved vs. linear) [109], etc., may have a significant impact on the estimation of muscle mass.

The most direct US measures of skeletal muscle are muscle thickness (i.e., the distance between muscle fascia), and CSA (largest cross-sectional area perpendicular to the fiber direction of the muscle), which can be measured directly bedside [110]. Muscle volume can be estimated using different prediction equations [111,112]. Muscle volume estimation may be more closely related to DXA-determined sarcopenia compared to muscle thickness measurement, which has been shown for the rectus femoris muscle of healthy older men and women [113]. In patients with cirrhosis, CSA measurement has shown better agreement with MRI than simple muscle thickness [105]. On the other hand, muscle thickness assessment is faster and easier to learn by clinicians. A recent study of hospitalized patients showed that using the average of only two measurements of rectus femoris or quadriceps thickness showed excellent reproducibility with no difference between experienced and unexperienced raters [114].

US offers the possibility of site-specific evaluation of muscle loss [115]. The most common sites for evaluation are the anterior thigh muscles [108,113,116], abdominal muscles [117], the psoas muscle [106,118], and the forearm muscles [108,116]. The forearm and anterior thigh muscles have previously been shown to be best correlated to DXA lean mass in patients with multi-organ failure and edema [119]. An interesting feature of the anterior thigh muscles is that they adapt fast to changes in loading, which makes them suitable for repeated measurements before and after interventions or periods of acute illness or immobilization [120,121].

There are few prognostic studies using US-determined muscle mass to predict adverse outcomes in patients with cirrhosis. One prospective study reported a higher risk of death and re-hospitalization in patients with low psoas muscle index [118]. A limitation of that study was that it was only possible to assess the psoas muscle index in 72% of the patients due to decompensation (edema, ascites). Another study reported an increased risk of death in patients with low abdominal muscle thickness [117]. However, muscle mass estimation based on abdominal muscle thickness may be falsely low in patients with ascites due to distension of the abdominal wall.

The major limitation to the implementation of US as a primary choice for skeletal muscle quantification is the challenge of standardization and reliability. This is also reflected by the current lack of diagnostic cut-offs, and no sarcopenia consensus groups recommend US for skeletal muscle assessment (Table 1). One group measured different variables related to skeletal muscle mass in patients with cirrhosis, including the Subjective Global Assessment, anthropometric measures, and ultrasound, in order to develop a simple method to diagnose sarcopenia [116]. They found that a nomogram based on BMI and the feather index (thickness of compressed/non-compressed muscle bulk) of the anterior thigh muscles was the best model for diagnosing sarcopenia when using L3-SMI (CT or MRI) as the reference standard. This study provides a reasonable approach to a bedside technique, which is easy to implement, but it remains to be validated in different populations and by clinicians with different US skills.

Currently, US is not ready for routine clinical evaluation of skeletal muscle. We recognize that, in the absence of other options, ultrasound may be a reasonable alternative, provided that local expertise and standardized methods for reliable, repeated assessments, as well as locally validated cut-offs, are present for clinically meaningful decision making.

## 4. Beyond Sarcopenia—Assessing Muscle Quality and Fat Distribution

Obesity is a well-known risk factor for multiple health-related problems. The combination of low muscle mass and obesity (sarcopenic obesity) is probably even worse, which has also been demonstrated in patients with liver disease [122,123]. The increasing number of patients with MASLD has prompted increased attention to this double pathological challenge.

Reports on advanced imaging assessment of both muscle and fat based on CT scans are increasing in numbers and provide a more nuanced picture of the fat compartment than simple BMI measurement. This includes analysis of fat infiltration in muscle (myosteatosis) as well as visceral and subcutaneous fat mass assessment. Myosteatosis and high visceral/subcutaneous fat ratio likely reflect an unfavorable metabolic and inflammatory condition, which distorts the homeostasis of the liver–muscle–fat axis by changing the balance of adipokines, myokines and pro-/anti-inflammatory cytokines. This speeds up liver disease progression, which again negatively affects muscle and fat metabolism (Figure 1).

All the non-invasive imaging modalities discussed in the previous section offer the possibility of fat measurement in different ways and with more or less evidence of a prognostic value in patients with liver disease.

CT is the best-studied modality. Intramuscular fat accumulation (myosteatosis) affects tissue radiodensity measured in HU, which can be used as a measure of skeletal muscle quality. The most commonly reported method is to perform semi-automated radiodensity measurements of skeletal muscles at the L3 level. In cohorts with mixed etiology of liver disease, myosteatosis is more prevalent than sarcopenia, and both sarcopenia and myosteatosis are associated with a worse prognosis [36,43,124,125,126,127]. Some studies have even found myosteatosis to be more closely correlated to strength measurements [38] and to be a better predictor of poor outcomes than muscle mass [59,128]. The degree of myosteatosis also seems to be closely linked to the degree of MASLD [129,130], and more closely associated with steatohepatitis and fibrosis than muscle mass in this patient group [131]. Further myosteatosis but not L3-SMI is associated with low cardiorespiratory fitness in patients with end-stage liver disease [132]. Cut-offs for myosteatosis in men and women have been proposed based on HU in patients with cirrhosis in a large retrospective cohort recently [127]. Prospective research studies are needed to validate these cut-offs and clarify the independent prognostic value of myosteatosis in different stages and etiologies of liver disease and possibly also in different ethnic groups. Early detection of myosteatosis may be important for the prevention of liver disease progression, not least in MASLD, since this pathological marker may appear before the loss of skeletal muscle mass [129,133].

Visceral obesity has also been associated with a poor prognosis in patients with liver disease, in particular those with MASLD. In a cohort of patients on the transplant list with mixed etiologies of liver disease, a high ratio of visceral to subcutaneous fat combined with sarcopenia displayed a mortality risk that by far superseded the mortality risk of having either sarcopenia alone or visceral obesity alone [134]. Likewise, too little subcutaneous fat has also been linked to increased mortality risk [135]. Cut-offs for pathological visceral and subcutaneous fat have, to our knowledge, not been validated sufficiently in patients with cirrhosis. A recent study published a nomogram incorporating both myosteatosis and the visceral-to-subcutaneous fat ratio together with L3-SMI, the model for end-stage liver disease (MELD), and the Child–Turcotte–Pugh score to predict 1-, 2-, and 3-year mortality with good discriminative power [136]. Models including multiple risk factors may be more accurate than single-risk-factor models. Automated image analysis and AI may in the future favor models including multiple body composition measures.

MRI has superior soft tissue resolution compared to CT, and it is therefore ideal for visualization and quantification of fat compartments. Results similar to CT have been reported with MRI, although there are fewer studies. Both myosteatosis [137,138,139] and a high visceral-to-subcutaneous fat ratio [14,140] seem to be superior in predicting adverse outcomes compared to skeletal muscle mass alone. On the contrary, one study did not find any association between myosteatosis on MRI and liver disease indices in patients with MASLD, which may be due to lower disease burden in that study population [15]. Abdominal scans [14,139,140], as well as thigh muscle [103,138], have been used to assess myosteatosis, but it is unknown which is the best predictor of poor metabolic health and adverse outcomes. The quadriceps muscle may be a more functionally relevant location to assess myosteatosis, but also requires additional expensive MRI scans. Protocols for faster whole-body MRIs with automated reports of body composition measures are currently being developed [141,142,143], and have also been tested in patients with MASLD [144]. MRI is thus a promising modality for quantitative body composition measurement if it becomes more widely accessible and validated in different liver disease populations.

DXA can reliably determine body fat mass including visceral fat estimation. To our knowledge, however, there are no dedicated studies investigating the prognostic value of measuring different body fat compartments with DXA in patients with cirrhosis.

BIA-determined fat mass index (BIA-determined fat mass normalized to height squared (BIA-FMI)) has been investigated as a screening tool for MASLD in different Asian populations with liver disease [145,146] and has shown high accuracy for predicting hepatic steatosis. However, one study has also shown that BIA-FMI is not better than simple BMI or waist circumference for this purpose [147], and thus the relevance of using BIA-FMI in patients with cirrhosis remains to be clarified. Phase angle is an indirect measure of cellular health and low phase angle could potentially reflect poor skeletal muscle quality, as well as unfavorable fat distribution and/or low skeletal muscle-to-fat mass ratio, but this is still unknown.

Ultrasonography is emerging as a technique to assess muscle quality. Echo intensity (EI) can be used to estimate fatty and fibrous infiltration in muscle, and shear wave velocity estimates tissue stiffness as a proxy for muscle quality. The methods are challenged by suboptimal reproducibility but reliability studies are emerging. Measurement of EI in the rectus femoris muscle has shown moderate test–retest reliability in a small cohort of patients with cirrhosis [105], and measurement of shear wave velocity in rectus femoris has shown high feasibility and intra-rater reproducibility in patients with cirrhosis [148]. Additionally, EI and shear wave velocity have shown moderate correlations to MRI-determined myosteatosis in healthy young and old adults, as well as in patients with non-small-cell lung cancer [149]. To the best of our knowledge, there are no studies investigating the prognostic impact of US-based muscle quality in patients with cirrhosis. The fast technological development of US equipment and AI may help to promote the necessary development of standardized acquisition protocols and increase the number of reliability studies (Table 3).

Collectively, the assessment of body fat compartments has the potential to improve prognostic models in patients with cirrhosis and may have equally or even superior prognostic significance compared to skeletal muscle mass assessment.

## 5. Summary and Perspectives

Table 2 and Table 3 summarize features and evidence of commonly applied advanced non-invasive methods to quantify body composition in patients with cirrhosis.

Advanced non-invasive quantitative assessment of skeletal muscle mass has good prognostic value and is relevant to confirm the diagnosis of sarcopenia in patients with cirrhosis. In some patients with, for example, acute illness or neuromuscular or skeletal comorbidities, it may even be the primary diagnostic test. L3-SMI based on CT scans is currently the best-validated imaging method for the quantification of skeletal muscle in patients with cirrhosis, and cut-offs are established for European/North American as well as Asian populations. DXA-ASMI and BIA-ASMI are reasonable radiation-friendly alternatives to CT that should be considered as the first choice for repeated assessments. Arm lean mass may be even more reproducible and accurate than ASMI in patients with cirrhosis due to the challenge of fluid retention in the legs of some patients. Future research should focus on the stratification of skeletal muscle cut-offs into risk categories based on CT, DXA, and BIA and differentiation according to liver disease severity, etiology, ethnicity, and gender. Quantification of skeletal muscle with US should currently focus on standardization and reliability studies followed by studies with hardcore endpoints to determine meaningful cut-offs.

Quantitative assessment of muscle quality and body fat compartments are good prognostic markers that may even be superior to the measurement of skeletal muscle mass. CT and MRI are currently the best-studied methods, but they require validation of cut-off values for myosteatosis and the visceral to subcutaneous fat ratio before clinical implementation is possible. US has the potential for inexpensive, easily accessible, and radiation-free repeated assessments of muscle quality of individual muscle groups. The clinical use is currently limited but standardization and reliability studies are encouraged. The prognostic value of DXA-determined fat distribution in patients with cirrhosis is lacking, and prospective studies are encouraged.

MRI is ideal for studying the pathophysiology of body composition changes in cirrhosis. Currently, clinical use is limited by low availability, but faster scan protocols, automation, and increased availability may make MRI superior for clinical body composition analysis in the near future.

There is good evidence that dietary and training interventions improve body composition in patients with liver disease, but little is known about whether these treatments improve hardcore endpoints such as survival. One study showed increased muscle mass together with increased survival and reduced hospitalization 3 years after a short strength training intervention of only 12 weeks [104]. Similar long-term interventional follow-up studies with repeated assessments of body composition and the relation to changes in prognosis are encouraged to improve knowledge about the time horizon for body composition changes and changes in prognosis.

The use of automated analysis introduces easy access to validated imaging biomarkers such as the CT-based L3-SMI, which are ready to be implemented in clinical routine. The current challenge is to provide an infrastructure, which supports automatic reporting of body composition biomarkers and the dissemination of knowledge about how to respond to abnormal values (medical, dietary, or training intervention). The development of AI can further help to identify combinations of multiple body composition measures that most reliably predict adverse outcomes.

Body composition measurement is highly relevant in patients with cirrhosis. Advanced non-invasive imaging biomarkers of the skeletal muscle and fat compartments may provide important input to traditional prognostic models like the MELD, which may improve the identification of patients at risk, target medical as well as nutritional and exercise therapy, and prioritize surgical treatment.

## Figures and Tables

**Figure 1 diagnostics-14-02191-f001:**
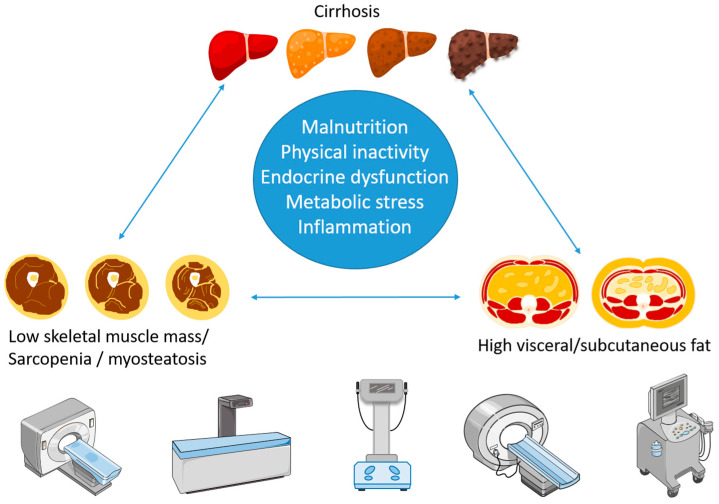
Illustration of the liver–muscle–fat cycle and different methods used to quantify skeletal muscle and fat.

**Table 1 diagnostics-14-02191-t001:** Recommendations for measurement of skeletal muscle mass in healthy adults and patients with liver disease.

Consensus Report	Modality for Skeletal Muscle Mass Assessment (Cut-Off)	Population
EWGSOP2 [23]	DXA or BIA (M: ASM < 7.0 kg/m^2^, W: ASM < 5.5 kg/m^2^) Alternatively, CT or MRI (no specified cut-offs)	Older adults
IWGS [21]	DXA (M: ALM/h < 7.23 kg/m^2^, W: ALM/h < 5.67 kg/m^2^)	Older adults
FNIH [22]	DXA (M: ALM/BMI < 0.789, W: ALM/BMI < 0.512) DXA (M: ALM < 19.75 kg, W: ALM < 15.02 kg)	Older adults
AWGS [8]	DXA (M: ASM < 7.0 kg/m^2^, W: ASM < 5.4 kg/m^2^) BIA (M: ASM < 7.0 kg/m^2^, W: ASM < 5.7 kg/m^2^)	Older adults
SDOC [24]	Recommends not to use DXA lean mass. No recommendations for other methods.	Older adults
JSH [11]	CT (M: L3-SMI < 42 kg/m^2^, W: L3-SMI < 38 kg/m^2^) BIA (M: ASM < 7.0 kg/m^2^, W: ASM < 5.7 kg/m^2^)	Liver disease
EASL [12]	CT (M: SMI < 50 kg/m^2^, W: SMI < 39 kg/m^2^) DXA (reference to cut-off for healthy populations) BIA (no specified cut-offs)	Liver disease
INASL [13]	CT (M: L3-SMI < 42 kg/m^2^, W: L3-SMI < 38 kg/m^2^) BIA (M: ASM < 7.0 kg/m^2^, W: ASM < 5.7 kg/m^2^) DXA (M: ASM < 7.0 kg/m^2^, W: ASM < 5.4 kg/m^2^)	Liver disease
AASLD [10]	CT (M: L3-SMI < 50 kg/m^2^, W: L3-SMI < 39 kg/m^2^)	Liver disease
ESPEN [25]	CT L3-SMI (no specified cut-offs) MRI L3-SMI (no specified cut-offs) If no overhydration, consider DXA or BIA (no specified cut-offs).	Liver disease Refers to EASL guidelines

IWGS: International Working Group on Sarcopenia, FNIH: the Foundation for the National Institutes of Health, EWGSOP2: the European Working Group of Sarcopenia in Older People, AWGS: the Asian Working Group on Sarcopenia, SDOC: the Sarcopenia Definition and Outcomes Consortium, JSH: the Japanese Society of Hepatology, EASL: the European Association for the Study of Liver Diseases, INASL: the Indian National Association for the Study of the Liver, AASLD: the American Association for the Study of Liver Diseases, ESPEN: European Society for Clinical Nutrition and Metabolism. DXA: Dual-energy X-ray Absorptiometry, BIA: Bioelectrical Impedance Analysis, CT: Computed Tomography, MRI: Magnetic Resonance Imaging, L3-SMI: skeletal muscle index at the L3 level, ASM: appendicular skeletal muscle, ALM: appendicular lean mass, h: height, BMI: body mass index, M: men, W: women.

**Table 2 diagnostics-14-02191-t002:** Features of commonly applied advanced non-invasive methods to quantify skeletal muscle and fat compartments.

Modality	Main Advantages	Main Disadvantages
CT	Validated cut-offs for sarcopenia. Possibility of detailed analysis of fat. Often available for other diagnostic purposes.	High radiation dose limits the possibility of repeated assessments.
DXA	High precision and relatively inexpensive.	Influenced by hydration (lower precision in decompensation or obesity)
BIA	Readily available, fast, and inexpensive.	Influenced by hydration Low precision
MRI	Best soft tissue resolution, which makes it suitable for detailed analysis of both muscle and fat compartments.	Limited access Expensive
US	Can be performed bedside and simultaneously with liver ultrasound diagnostics.	No standardization Limited reproducibility

CT: Computed tomography, DXA: Dual-energy X-ray Absorptiometry, BIA: Bioelectrical Impedance Analysis, MRI: Magnetic Resonance Imaging, US: Ultrasonography.

**Table 3 diagnostics-14-02191-t003:** Evidence for the prognostic value of different body composition measures and proposed research directions.

Modality	Measures	Cut-Offs/ Evidence	Proposed Research
CT	L3-SMI Myosteatosis VSR	Yes/high Yes/medium No/low	Stratification and differentiation of cut-offs Validation, stratification, and differentiation of cut-offs Establishment of cut-offs Automated reporting and implementation in advanced models for risk prediction
DXA	ASMI or ALM Arm lean mass Fat mass	Yes/low Yes/medium No/low	Validation, stratification, and differentiation of cut-offs Validation, stratification, and differentiation of cut-offs Establishment of prognostic value and cut-offs Intervention studies with repeated assessments
BIA	ASMI or ALM Arm lean mass Fat mass Phase angle	Yes/low Yes/medium No/low Yes/medium	Validation, stratification, and differentiation of cut-offs Validation, stratification, and differentiation of cut-offs Establishment of prognostic value and cut-offs Validation, stratification, and differentiation of cut-offs Intervention studies with repeated assessments
MRI	L3-SMI Myosteatosis VSR	Yes/medium No/medium No/low	Validation, stratification, and differentiation of cut-offs Establishment and validation of cut-offs Establishment and validation of cut-offs Pathophysiology of body composition changes in liver disease Automated reporting and AI analysis to improve risk prediction
US	Muscle thickness, CSA, and volume, Pennation angle, Echo intensity, Feather index, Shear wave velocity	No/low	Standardization and reliability studies Establishment of prognostic values and cut-offs Intervention studies with repeated assessments

CT: Computed Tomography, DXA: Dual-energy X-ray Absorptiometry, BIA: Bioelectrical Impedance Analysis, MRI: Magnetic Resonance Imaging, US: Ultrasonography, L3-SMI: skeletal muscle index at the L3 level. VSR: visceral-to-subcutaneous fat ratio. ASMI: appendicular skeletal muscle index. ALM: appendicular lean mass. Validation, i.e., establishment of cut-offs or testing the ability of established cut-offs to predict morbidity and mortality. Stratification and differentiation, i.e., establishment of cut-offs for different severity categories of sarcopenia, different etiologies of liver disease, or according to gender.

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
