# Peer review of "Quantitative Assessment of Body Composition in Cirrhosis"

_diagnostics, 2024, doi:10.3390/diagnostics14192191_

Round 1

Reviewer 1 Report

Comments and Suggestions for Authors

This narrative review explores the significance and current state of advanced, non-invasive methodssuch as CT, DXA, BIA, MRI, and ultrasoundfor quantifying body composition, including skeletal muscle mass, sarcopenic obesity, myosteatosis, and fat distribution, in patients with liver disease.

My Comments:

1. The title, "Quantitative assessment of body composition in liver disease," suggests a comprehensive discussion on body composition across various chronic liver diseases. However, the content primarily focuses on cirrhosis, with limited coverage of other chronic liver conditions. To enhance the scope and differentiate this review from existing literature on similar topics, I recommend categorizing and discussing body composition assessments across different types of chronic liver diseases (e.g., hepatitis, NAFLD, cirrhosis, HCC, etc.) to provide a more balanced and comprehensive review.

2. There are several minor errors in the manuscript that need to be addressed. For example, phrases such as "International consensus groups have established optimal populations based cut-offs based on mortality data (Table 1)" and "Only few studies have been unable to find association between L3-SMI and mortality in patients with less severe liver disease" require modification. Additionally, some abbreviations should be accompanied by their full forms upon first mention to ensure readability and understanding for a wider audience.

3. In the section discussing muscle mass, sentences such as Other options are to measure pennation angle (angle between the muscle fibers and deep aponeurosis), echo intensity (reflects muscle quality, i.e., fat and connective tissue infiltration), and feather index (measuring thickness of both compressed and non-compressed muscle bulk). Recently, shear wave velocity of skeletal muscle has also been studied in patients with cirrhosis as a measure of muscle quality, appear disorganized and blend multiple aspects of muscle assessment. It would be more effective to separate discussions on muscle mass from those on muscle quality and other body composition parameters. This will improve the structural clarity and focus of the review.

Comments on the Quality of English Language

Minor editing of English language required.

Author Response

Comment 1: The title, "Quantitative assessment of body composition in liver disease," suggests a comprehensive discussion on body composition across various chronic liver diseases. However, the content primarily focuses on cirrhosis, with limited coverage of other chronic liver conditions. To enhance the scope and differentiate this review from existing literature on similar topics, I recommend categorizing and discussing body composition assessments across different types of chronic liver diseases (e.g., hepatitis, NAFLD, cirrhosis, HCC, etc.) to provide a more balanced and comprehensive review.

Response 1: We agree with the Reviewer that “….. in liver disease” may be a too broad term in relation to the content of the review. We have accordingly changed the title to “….. in cirrhosis” instead. 

We agree with the Reviewer that a broader discussion of body composition assessment across different types of chronic liver disease could be an interesting focus of a future review.

It is our impression that most studies include patients with mixed etiologies of liver disease, except for studies on NAFLD/MASLD. Future studies could focus more on stratification and differentiation of cut-offs, which we have proposed in table 2.

Comment 2: There are several minor errors in the manuscript that need to be addressed. For example, phrases such as "International consensus groups have established optimal populations based cut-offs based on mortality data (Table 1)" and "Only few studies have been unable to find association between L3-SMI and mortality in patients with less severe liver disease" require modification. Additionally, some abbreviations should be accompanied by their full forms upon first mention to ensure readability and understanding for a wider audience.

Response 2: We thank the Reviewer for pointing out these errors, which has improved the manuscript considerably. We have reviewed the manuscript focusing on these errors.

Comment 3: In the section discussing muscle mass, sentences such as “Other options are to measure pennation angle (angle between the muscle fibers and deep aponeurosis), echo intensity (reflects muscle quality, i.e., fat and connective tissue infiltration), and feather index (measuring thickness of both compressed and non-compressed muscle bulk). Recently, shear wave velocity of skeletal muscle has also been studied in patients with cirrhosis as a measure of muscle quality,” appear disorganized and blend multiple aspects of muscle assessment. It would be more effective to separate discussions on muscle mass from those on muscle quality and other body composition parameters. This will improve the structural clarity and focus of the review.

Response 3: We thank the Reviewer for the comment. We agree that this sentence is misplaced and have deleted it from the manuscript.

Reviewer 2 Report

Comments and Suggestions for Authors

The manuscript provides a comprehensive overview of non-invasive techniques for assessing body composition in liver disease. It might benefit from a more distinct section outlining the clinical implications of these assessments—how do they influence patient management, and are there any treatment modifications based on these assessments?

The manuscript references several guidelines but could expand on how these imaging techniques align with or differ from current clinical guidelines in gastroenterology and hepatology. This would help clinicians understand how to integrate this knowledge into practice. Meanwhile, it could further specify potential studies that might address existing gaps in the literature, particularly in correlating imaging findings with clinical outcomes in liver disease.

While Figure 1 is mentioned, the manuscript could benefit from additional visual aids, such as flowcharts or graphs comparing the sensitivity and specificity of the different techniques or depicting the progression of body composition changes in liver disease Mention is made of artificial intelligence and its potential impact on imaging analysis. A deeper dive into how AI is currently being integrated and its future applications would be valuable, considering the rapid evolution of technology in medical imaging.

Comments on the Quality of English Language

Moderate editing of English language required.

Author Response

Comments and Suggestions for Authors

Comment 1: The manuscript provides a comprehensive overview of non-invasive techniques for assessing body composition in liver disease. It might benefit from a more distinct section outlining the clinical implications of these assessments—how do they influence patient management, and are there any treatment modifications based on these assessments?

Response 1: We agree with the Reviewer that the clinical implications of body composition assessment are relevant to address, and we have accordingly included a brief overview in the introduction (p. 1).  It is our opinion that a distinct section about clinical implications of body composition assessment in cirrhosis is beyond the scope of the present review.

Comment 2: The manuscript references several guidelines but could expand on how these imaging techniques align with or differ from current clinical guidelines in gastroenterology and hepatology. This would help clinicians understand how to integrate this knowledge into practice. Meanwhile, it could further specify potential studies that might address existing gaps in the literature, particularly in correlating imaging findings with clinical outcomes in liver disease.

Response 2: We agree with the Reviewer that the differences between sarcopenia guidelines for healthy adults and guidelines in gastroenterology and hepatology could be better addressed. We have therefore to some extent discussed this in the section about DXA (p. 7). We have tried to further emphasize this in the section about muscle mass measurement with CT (p. 5).

There is, however, to our knowledge no clinical guidelines on advanced body fat composition assessment.

We have made recommendations about potential future studies, which partly build on the current lack of knowledge in the current guidelines (table 2).

Comment 3: 

While Figure 1 is mentioned, the manuscript could benefit from additional visual aids, such as flowcharts or graphs comparing the sensitivity and specificity of the different techniques or depicting the progression of body composition changes in liver disease Mention is made of artificial intelligence and its potential impact on imaging analysis. A deeper dive into how AI is currently being integrated and its future applications would be valuable, considering the rapid evolution of technology in medical imaging.

Response 3: The Reviewer has a point, but sensitivities and specificities of the individual techniques vary a lot from study to study, not least depending on the comparator. Providing valid percentages would require a dedicated meta-analysis, which is beyond the scope of the present review.

 We acknowledge that visualization of body composition changes over the course of disease progression would give be a good aid to the reader. However, we are not in the possession of copyrighted images from all the different modalities, whilst we are not able to publish such images in our review.

We agree that a deeper dig into the clinical potential of AI in body composition analysis would be a valuable scientific contribution. However, we find it is beyond the scope of the present review.

Round 2

Reviewer 1 Report

Comments and Suggestions for Authors

This title adjustment feels more like a superficial change rather than a substantive improvement in the scope of the review. Existing reviews on body composition and cirrhosis, such as PMID: 32621329 and PMID: 37095222, already cover similar ground, which diminishes the novelty and overall value of this manuscript. To enhance its contribution to the literature, I strongly recommend that the authors expand the discussion to cover different types of chronic liver diseases, such as hepatitis B/C, alcoholic liver disease, and non-alcoholic fatty liver disease (NAFLD). A more comprehensive review would increase the relevance and uniqueness of the manuscript.

Comments on the Quality of English Language

none

Author Response

Comment 1: This title adjustment feels more like a superficial change rather than a substantive improvement in the scope of the review. Existing reviews on body composition and cirrhosis, such as PMID: 32621329 and PMID: 37095222, already cover similar ground, which diminishes the novelty and overall value of this manuscript. To enhance its contribution to the literature, I strongly recommend that the authors expand the discussion to cover different types of chronic liver diseases, such as hepatitis B/C, alcoholic liver disease, and non-alcoholic fatty liver disease (NAFLD). A more comprehensive review would increase the relevance and uniqueness of the manuscript.

Response 1: Answer: We are aware that others have reviewed the relevance of body composition assessment in patients with cirrhosis. We also agree with the reviewer that a discussion of body composition assessment in patients with different etiologies of liver disease would be a valuable new contribution to the scientific community. We encourage more original research studies that stratify according to etiology of cirrhosis, but are currently unable to submit a manuscript that covers this topic, partly due to a scarcity of original studies.

However, we still find our contribution relevant given the fast technological development, and the importance of improving understanding and encouraging development of strategies to prevent or treat a pathological body composition in patients with cirrhosis.

In contrast to the mentioned reviews published by others (PMID: 32621329 and PMID: 37095222), we include a discussion of MRI and US for body composition assessment. We also provide a more detailed analysis about the most recent evidence of DXA and BIA for body composition assessment, acknowledging the fact that several different methods are applicable for body composition assessment, and that the choice of method also depends on local availability of equipment and diagnostic traditions.

Reviewer 2 Report

Comments and Suggestions for Authors

significant improvement after revising

one more minor point

It is mentioned that recent clinical practice guidelines recommend the early diagnosis of sarcopenia. Including a statistical analysis of how early diagnosis impacts clinical outcomes could provide strong support for these recommendations.

Comments on the Quality of English Language

high quality

Author Response

Comment 1: It is mentioned that recent clinical practice guidelines recommend the early diagnosis of sarcopenia. Including a statistical analysis of how early diagnosis impacts clinical outcomes could provide strong support for these recommendations.

Response 1: We agree that a statistical analysis of the clinical implications of early diagnosis of sarcopenia is relevant to support this claim. We are, however, not familiar with such an analysis and do not have data to support the claim ourselves. We merely refer to practice guidelines for sarcopenia assessment in cirrhosis.